nanotechnology

superabsorbent, sodium polyacrylate, micropump, solution processing, disposable device, switchable valve

**Author for correspondence:**
Gokul Chandra Biswas
e-mail: gcbiswas-geb@sust.edu

[†]Present address: School of Life Sciences, Shahjalal University of Science and Technology, Sylhet 3114, Bangladesh.

# A simple micropump based on a freeze-dried superabsorbent polymer for multiplex solution processing in disposable devices

Gokul Chandra Biswas[1,†], Md. Mohosin Rana[1], Takekoshi Kazuhiro[2] and Hiroaki Suzuki[1]

[1]Graduate School of Pure and Applied Sciences, University of Tsukuba, 1-1-1 Tennodai, Tsukuba, Ibaraki 305-8573, Japan
[2]Faculty of Medicine, University of Tsukuba, 1-1-1 Tennodai, Tsukuba, Ibaraki 305-8675, Japan

GCB, 0000-0001-9097-6426

We describe a simple micropump for disposable microfluidic devices. The pump is constructed using a freeze-dried disc of a superabsorbent polymer (SAP). The disc absorbs a solution in a flow channel and swells upward in a pumping chamber. Despite the simple structure of this device, the rate of absorption remains constant and can be adjusted by changing the composition of the SAP, its size, the dimensions of the flow channel and the medium to be absorbed. The pumping action can be initiated by applying an electrical signal using a switchable hydrophobic valve. The integrated approach of the SAP pump and switchable valve could facilitate the automatic processing of many solutions required for bioassay.

## 1. Introduction

Micropumps are indispensable devices in the processing of solutions in microfluidic systems. Ideally, micropumps should be able to operate with minimal manual intervention [1,2]. To achieve this goal, the structure, fabrication and operation of each component should be as simple as possible. Although micropumps of various designs and principles have been proposed, active pumps, such as those based on pneumatic and electroosmotic principles, need additional power or pressure sources [3,4]. In this respect, passive solution transport based on capillary action provides a realistic alternative operating mechanism [5,6]. The capillary force-driven pumping is spontaneous and suitable for integration in many microfluidic systems, but requires aggregation of various capillary

**Figure 1.** Absorption of an aqueous solution by SP. (*a*) Random coil of polymer chains in the dry state. (*b*) Uncoiling of the polymer chains in the wet state and absorption of water molecules.

micro-patterns in a confined pumping unit and the solution volume should be predetermined. Passive solution flow by manipulating surface energy in the liquid drop is also quite simple and attention-grabbing [7], but dependent on the large volume and shape/size of the drop. The transport of small volumes of solution by created evaporation-coupled capillary pressure difference on the solution meniscus at the outlet has also been reported [8]. The pressure adjustment on the liquid meniscus requires the maintenance of constant temperature and relative humidity, and the handling of high volume of solution has not been addressed. Another method, vacuum suction-based degassing of permeable PDMS (poly(dimethylsiloxane)) has been reported as a potential tool for passive flow without bubbling [9]. Nevertheless, the system entails the long exposure of low pressure conditions to gain the fluid driving force. The pumping with the above-mentioned passive principles is self-directed and appealing in terms of their portability, handling of small dead volume of solutions in glass/PDMS-based devices. However, the on-device integration of these pumping components with the flow system may require additional fabrication steps. Moreover, with only the capillary action, the ability of a micropump to process solutions is limited.

In that sense, the passive fluid delivery using absorption by materials such as paper [10–12], threads [13], cotton [14,15], sponge [16] or nitrocellulose membrane [17] provides a realistic solution. In particular, paper-based devices have attracted attention due to their cost effectiveness, simple fabrication and convenience in handling. However, the low absorbing capacity per unit area of paper makes the integration of paper-based micropumps with other microfluidic components for multiple processing of solutions impractical [12,18]. The paper-based devices are superior when the solutions move rapidly. In cases where low flow velocity or sufficiently long reaction time is needed and/or the volume of solutions is relatively large, a large (specific) surface area will be required by the paper materials. This problem can be solved by using a superabsorbent polymer (SAP) [18,19], which is known to absorb and hold distilled water 10–1000 times its dry weight [20,21]. The SAP, along with paper material that shows the risk of back-flow of solution, could assist in running the multi-step (i.e. coating, blocking, washing, immobilization, conjugation, etc.) flow of many solutions in the paper-based assay process with its high absorption properties [18].

Due to the interest in their wide-spread applications (e.g. home decoration, sanitation, agriculture, drug delivery to diagnostic assay), the majority of the earlier reports discussed the material characteristics, swelling nature, water-absorbing phenomena and synthesis of SAPs [20–23]. Still, the microfluidic applications of SAPs are few. Previously, the liquid absorbing phenomena of SAP were used for sweat [24] and waste collection [25]. Recently, the swelling behaviour of the SAP has been used as a valve to regulate sweat flow in skin-mounted devices [26]. Lee & Hsieh of National Chung Cheng University performed microfluidic cytometry using the superabsorbent bed of Carnation sanitary napkin and characterized the flow rate on the basis of channel dimensions [19]. However, in the case of fixed flow dimension, flow control was not stated. Alternatively, the use of powdery absorbent may raise the risk of blocking the flow path by its uncontrolled swelling property. Until now, the absorption/transport compatibility of the SAP to various types of solutions (especially clinically relevant fluids, e.g. blood, serum, protein, urine, etc.) in micro flow channels has not been investigated. Also, a great scope remains for the SAP to process multiple solutions in microfluidic devices with its high absorption capacity. In addition, the planar absorbent (e.g. paper or polymer) occupies more space in a system, which compromises the miniaturization [12]. Therefore, the compact solid-like structure of SAP that occupies vertical space could present potential benefits as a pump for integration in miniaturized platforms.

In this study, to address the limitations of absorbent-driven pumps, we propose an SAP-based passive pump for versatile processing of solutions. We used sodium polyacrylate (SP, $[-CH_2-CH(CO_2Na)-]_n$). SP is a representative SAP, which is used for commercial products such as disposable diapers [22,23,27]. Dry SP consists of random coils of polymer chains (figure 1*a*). When hydrated, the Na$^+$ ions detach from the

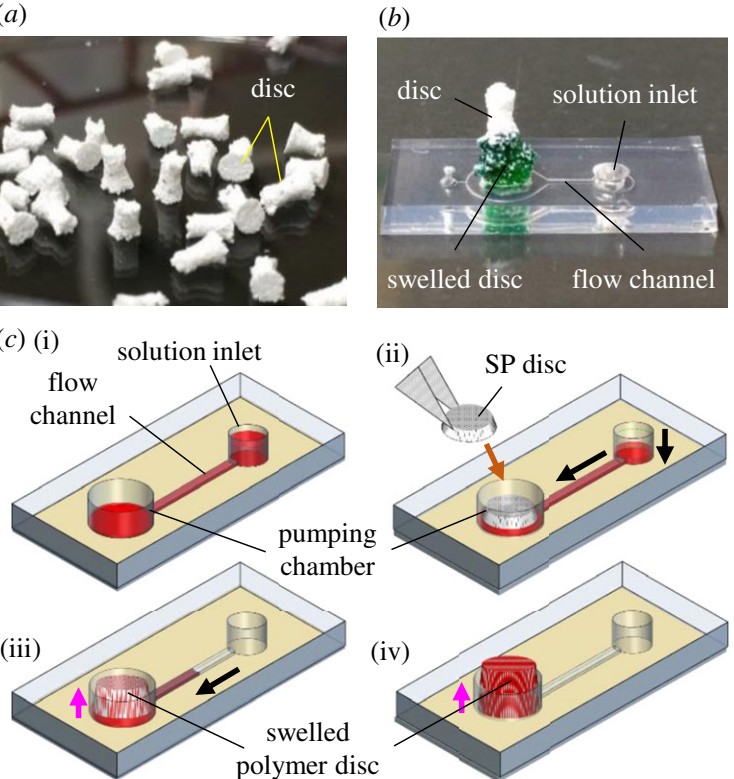

**Figure 2.** (*a*) Freeze-dried SP discs. (*b*) A device with an SP disc for pumping. A green food dye solution was used to visualize the swelling of SP. (*c*) Operation of the pump. (i) A solution is introduced. (ii) The SP disc is put into the pumping chamber. Pumping starts. (iii,iv) The solution is absorbed into the disc. Black and pink arrows indicate the direction of the solution flow and the swelling of the disc, respectively.

coils, producing negatively charged $COO^-$ groups (figure 1*b*) which repel each other and attract water molecules. To assemble a compact pump using this material, cylindrical discs were batch-fabricated using a mould (figure 2*a*; electronic supplementary material, figure S1). A unique feature of the disc is that it mainly swells in the vertical direction, which facilitates the integration of many micropumps in a limited space. Furthermore, to our surprise, a fairly constant rate of absorption was sustained over a period of hours with this simple, compact and inexpensive material, which opens up new possibilities for flow control on a disposable device.

# 2. Material and methods

## 2.1. Reagents and materials

Materials and reagents used for the fabrication and characterization of the devices were purchased from the following commercial sources: glass wafers (no. 7740, 3 inch diameter, 500 μm thick) from Corning Japan (Tokyo, Japan); a thick-film photoresist SU-8 25 from MicroChem (Newton, MA, USA) and a prepolymer solution of poly(dimethylsiloxane) (PDMS) (KE-1300T) from Shin-Etsu Chemical (Tokyo, Japan); potassium chloride (KCl), cellulose acetate (CA), bovine serum albumin (BSA), Tween 20, fluorescein, 1-hexanethiol and the dye sunset yellow FCF from Wako Pure Chemical Industries (Osaka, Japan); cross-linked SP from Sigma-Aldrich (St Louis, USA); the dyes Orange G and methyl violet from Merck (Darmstadt, Germany); food dyes from Kyoritsu Foods (Tokyo, Japan); cellulose filter paper (Whatman filter paper, Grade 2, no. 1001 150) from Whatman International (Maidstone, England); cotton thread (0.5 mm diameter) from S. Alam Product (Bangladesh); tissue paper from Basundhara Group (Bangladesh); cotton from Royal Trading (Bangladesh); and sponge from Office Supply BD (Bangladesh). All other reagents not mentioned here were obtained from Wako Pure Chemical Industries (Osaka, Japan). Milli-Q pure water (resistivity: 18 MΩ·cm) was used for the preparation of all solutions.

## 2.2. Device fabrication

The devices used in the experiments were fabricated by stacking a PDMS substrate with microfluidic structures onto a glass substrate. The thick-film photoresist (SU-8 25) was used to form a mould for the PDMS structures. The height of the flow channel was 50 μm for all devices, except for the one used for high rates of absorption. In this case, the height was 500 μm. For the device shown in figure 2, the length and width of the flow channels were 4.5 mm and 500 μm, respectively. An inlet with a diameter of 2.5 mm and an outlet with a diameter of 3 mm were made at the ends of the flow channel by punching (no. 1256, Takashiba Gimune Seisakujo, Hyogo, Japan). The device for the processing of multiple solutions, shown below, had 50 μm wide capillary valves at the entrance of the pumping chamber. The diameter of the pumping chamber was 5 mm, and the width of the flow channels was 200 μm. In all devices, 30 μm wide air vents were also added. The external ends of the air vents consisted of though-holes in the PDMS layer obtained using a disposable biopsy punch (Ref. BP-10F, Kai Industries, Seki, Gifu, Japan). The glass and PDMS substrates were attached by applying a slight pressure by hand.

## 2.3. Formation of SAP discs

Electronic supplementary material, figure S1 shows the steps required for the fabrication of the absorbing disc consisting of SP and CA particles. The mould was prepared by making through-holes in a 5-mm-thick PDMS substrate using a 3 mm diameter puncher (no. 1256, Takashiba Gimune Seisakujo, Hyogo, Japan). The PDMS mould was then placed on a glass substrate. First, SP powder (100 mg) was put into a tube, then water (600 μl) was added and the powder was stirred. 100 mg of CA particles were then added and stirred using a toothpick (electronic supplementary material, figure S1(i)). Again, 200 μl of water and 50 mg of CA particles were added and stirred.

This process produced a sticky semi-solid composite. The composite was put into the mould using a wet toothpick and pressed to make it compact. The mould with the composite was frozen for 2 h at −80°C and freeze-dried for 3.5 h at −50°C. The discs were removed from the mould by gently pushing them using a dry toothpick. The discs were stored in sealed containers to avoid contact with humidity in air. This process produced cylindrical tapered SP discs. The diameters at the bottom and at the top of the disc were 3 mm and 2.5 mm, respectively (figure 2a; electronic supplementary material, figure S1(iv)). The tapered shape promotes the upward swelling of the polymer in the pumping chamber.

## 2.4. Formation of the microvalve

The hydrophobic valves used in the device for automatic solution processing were formed following the procedure described previously [28]. A hydrophobic self-assembled monolayer (SAM) of 1-hexanethiol was formed on a platinum electrode. The electrode pattern was formed by sputtering and lift-off. The positive photoresist was used to delineate the active area of the electrode. Before the formation of the SAM, the electrode was cleaned in a solution containing water, 25% $NH_3$ and 30% $H_2O_2$ in a 4:1:1 volume ratio for 1 h at room temperature. The electrode was cleaned further in a 0.1 M KCl solution by potential cycling (10 times) in a potential range between −1.0 and 1.0 V (versus Ag/AgCl/sat. KCl). After drying with nitrogen gas, the cleaned electrode was immediately immersed in a 1 mM 1-hexanethiol aqueous solution containing 0.1 M ethanol for 150 s. After the SAM formation, the surface was rinsed again with pure ethanol and water. Finally, the device was dried by blowing nitrogen gas and was stored in a sealed container. When in use, the valve was opened by applying −1.0 V to the platinum-SAM valve electrode with respect to an Ag/AgCl electrode (2080 A-06T, Horiba, Kyoto, Japan) in a 0.1 M KCl solution. The valve could also be opened using a silver wire with AgCl.

## 2.5. Operation of the micropump

The pumping action of the disc was examined using a device with a single straight flow channel (flow dimension = length × width × height = 4.5 mm × 500 μm × 50 μm) (figure 2b; electronic supplementary material). The device operating procedure is shown in figure 2c (electronic supplementary material, movie S1). A solution was first introduced into the flow channel. The SP disc was then introduced into a pumping chamber with a through-hole of 3 mm diameter. The interior of the pumping chamber has a diameter of 4.5 mm, which leaves some free space around the disc (figure 2b; electronic supplementary material, figure S2). Without this space, the swelled disc can obstruct the flow channel exit, with a

consequent interruption in the absorption of the solution. Interestingly, the disc was found to expand mainly in the vertical direction even after a substantial volume was absorbed (figure 2b).

# 3. Results and discussion

## 3.1. Formation of the SAP disc

To accelerate absorption and maximize the volume of water absorbed, the discs must be very porous, and, to enhance the porosity, we used freeze-drying. However, it was virtually impossible to obtain SP discs with the desired shape. Furthermore, the discs showed a tendency to swell with consequent unpredictable changes in their shape; this can also cause the PDMS and glass substrates (which constitute the pump) to detach. To solve these problems, we added CA particles as a support (electronic supplementary material, figure S1) [29]. With this polymer composite, discs of various shapes could be obtained (electronic supplementary material, figure S1).

## 3.2. Characterization of the pump

The porous freeze-dried disc promotes the permeation of the solutions (figure 2a,b; electronic supplementary material, figure S1(iv)). We first used water to examine the influence of the SP/CA ratio on the average rate of absorption $A_r = V_s/t_a$, where $V_s$ is the total volume of solution absorbed and $t_a$ is the time needed to absorb the solution (electronic supplementary material). $A_r$ was found to be largest when the SP/CA ratio is 2/3 (electronic supplementary material, figure S3). Unless otherwise noted, we used this ratio in the rest of this work. We then determined $A_r$ by measuring the displacement of a meniscus in a tube connected to the device during water absorption (electronic supplementary material). With a disc of 5 mg weight, $A_r$ was found to be constant, except during the initial stage (figure 3a, inset). Considering the nature of the material used and the simple structure of the device, stable pumping at a constant $A_r$ is a surprising result. The initial higher $A_r$ may be caused by water in the pumping chamber diffusing through areas not occupied by the disc. A constant $A_r$ at all times might be realized by further optimizing the device design.

We also considered the absorption of aqueous solutions, including 0.1 M KCl, 0.1% (v/v) Tween 20 (surfactant), 0.5% (w/v) BSA, saliva, serum, whole blood, urine, phosphate buffered saline (PBS) and 0.9% NaCl (figure 3a; electronic supplementary material). The absorption capacity $A_c$ is given by the ratio between the weight of the maximum volume of liquid absorbed and the weight of the SP before absorption, i.e. $A_c = (W_1 - W_0)/W_0$, where $W_0$ and $W_1$ are the weights of the SP before and after absorption. We measured the largest value of $A_c$ for distilled water and the lowest value for the NaCl solution among non-viscous solutions, in agreement with previous reports [23]. The low absorption capacity of the NaCl solution is explained by the polyelectrolyte effect: the cations in the solution bind to the $COO^-$ groups of the SP and neutralize the polyanions. As the ionic strength of the solution increases, the osmotic pressure decreases and absorption slows down [22,23,27]. In figure 3a, we also show the values of $A_r$ obtained with different solutions using the device shown in figure 2. The dependence of $A_r$ on the nature of the solution was found to mirror that of $A_c$. As seen in the cases of BSA, saliva, serum and whole blood, viscosity also affects the absorption. $A_r$ and $A_c$ were particularly small with serum and whole blood. However, the most viscous solutions have a value of $A_r$ that is smaller compared with the change of $A_c$ as a function of solution type. For some solutions with low $A_r$ and $A_c$ values, it is not effective to absorb them directly into the SAP disc. A good alternative is to use water as a working medium, using its absorption into the SAP to draw another solution that follows. The water and the solution can be either connected or separated with air (electronic supplementary material, figure S4A). With this method, approximately a two-fold increase in $A_r$ was observed with whole blood and serum (electronic supplementary material, figure S4B).

The results shown in figure 3 indicate that $A_r$ of the pump can be adjusted by selecting an appropriate solution as a working medium and adjusting its concentration (electronic supplementary material, figure S5). Also, $A_r$ can be adjusted by changing the dimensions of the flow channel (flow resistance), the contact area of the SAP with the solution, and the size of the disc, as well as the SP/CA ratio. $A_r$ was found to increase to 250 nl s$^{-1}$ when the cross-section of the flow channel and the disc size were increased to 500 μm × 500 μm and 35 mg, respectively (electronic supplementary material, figure S6A). In applications like cell culturing or continuous monitoring of analytes, continuous supply of nutrient media or specific reagents during a period of hours or days is required [8,14]. With a small cross-section flow channel (200 μm wide, 50 μm high) and an SP/CA ratio of 1/3, 120 μl of water can be pumped for over 1.5 h (with $A_r$ of 0.02 μl s$^{-1}$), even with a disc of 5 mg weight (electronic supplementary material, movie

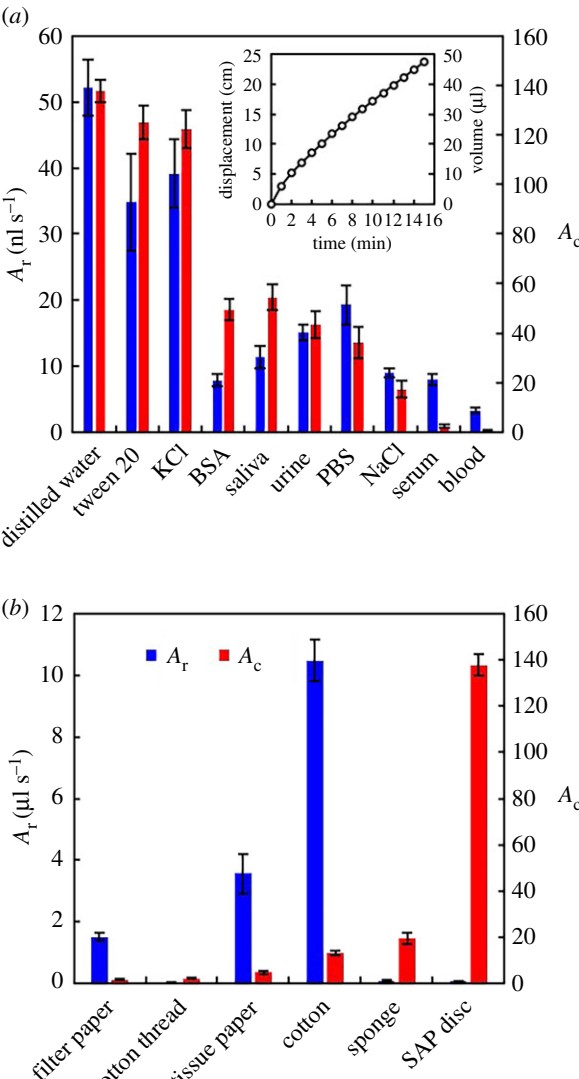

**Figure 3.** (a) Capacity of absorption ($A_c$) and average rate of absorption ($A_r$) of an SP disc. The inset shows the time evolution of the water meniscus displacement caused by the SP disc swelling. (b) $A_r$ and $A_c$ measured with various adsorbing materials. The blue and red bars represent $A_r$ and $A_c$, respectively. Error bars represent the standard deviation (number of replication, $n = 3$).

S2 and figure S6B). The $A_r$ can also be tuned by changing the temperature, and the absorption of water by SP has a maximum at around 55°C [23].

Compared to the $A_c$ values we measured for other materials (figure 3b), the SAP disc showed remarkably large $A_c$. On the other hand, $A_r$ of cotton was much higher than that of the SAP discs. Materials with a large $A_r$ will be suitable for rapid solution transport, whereas those with a large $A_c$ will be effective for moving solutions very slowly or holding them after use. We also compared the on-device absorption of nearly the same volume (approx. 15 mm³) of paper and SAP disc. The SAP disc transported approximately 23 times more solution than the 40-fold tissue paper (that resembled approximately 2 mm batch-fabricated paper material) (electronic supplementary material, figure S7A). Unlike the solution locking property of SAP disc, paper presented the issue of reverse-flow upon reaching the maximum absorption point (electronic supplementary material, figure S7B).

The SAP discs could also be stored for more than one month in an air-tight container conserving the performance as freshly prepared ones. On the other hand, $A_c$ and $A_r$ decreased by approximately 10% when the discs were stored in the open-air condition at room temperature for 30 days. Even in this case, deterioration of the performance was not significant. The result indicates that the SAP pumps are reproducible if they are stored properly.

With our pump, many problems that are encountered in other passive pumps were solved. Table 1 summarizes the problems and improvements made in our pump.

**Table 1.** Problems with other passive pumps and improvements made in the SAP disc pump.

| types | current passive pumps / problems | SAP disc pump / improvements |
|---|---|---|
| capillary force-driven pump [5,6] | — a construction is needed in pumping unit<br>— it is not suitable to withdraw solutions of large volumes | — no special structure is needed<br>— solutions of relatively large volumes can be withdrawn |
| surface tension-driven pump [7] | — intermediate fluid is necessary to move the solution to be transported<br>— a liquid droplet of a relatively large volume is necessary | — no intermediate fluid is required<br>— no droplet is necessary |
| evaporation-coupled capillary pump [8] | — performance is influenced by temperature and relative humidity<br>— it is not suitable to withdraw solutions of large volumes | — performance is not influenced by temperature and relative humidity<br>— solutions of relatively large volumes can be withdrawn |
| vacuum suction pump [9] | — fabrication is relatively complicated<br>— long exposure to low pressure is required to generate driving force | — fabrication is simple<br>— no such requirement is needed |
| absorption-based pumps (e.g. paper [10–12], threads [13], cotton [14,15], sponge [16] or nitrocellulose membrane [17]) | — absorbing capacity per unit area is low<br>— a relatively wide area is needed<br>— it is difficult to achieve slow pumping for a long time<br>— integration of many pumps is difficult | — high absorbing capacity per unit area can be achieved<br>— the structure is compact and accommodated in a small space<br>— solutions can be withdrawn slowly for a long time<br>— the simple structure facilitates integration of multiple pumps |
| SAP-fiber bed pump (Napkin bed) [19] | — the bed structure needs a space<br>— adjustment of flow rate is not easy<br>— pumping of clinically important fluids (i.e. blood, serum, protein, urine, etc.) has not been reported | — only a small space is sufficient with the vertically long disc<br>— flow rate can be adjusted by changing the SP/CA ratio, dimensions of the flow channel, size of the SAP disc, and salt concentration<br>— many clinically important fluids can be withdrawn |
| SAP powder based pump [18] | — there is a risk of blocking the flow channel due to its uncontrolled swelling behaviour<br>— the device may be contaminated with floating powder<br>— it is not suitable for multiplex and automatic processing of solutions | — there is no such risk because the disc swells upward in a controlled manner<br>— there is no risk of contamination<br>— the simple structure is suitable for multiplex and automatic processing of solutions |

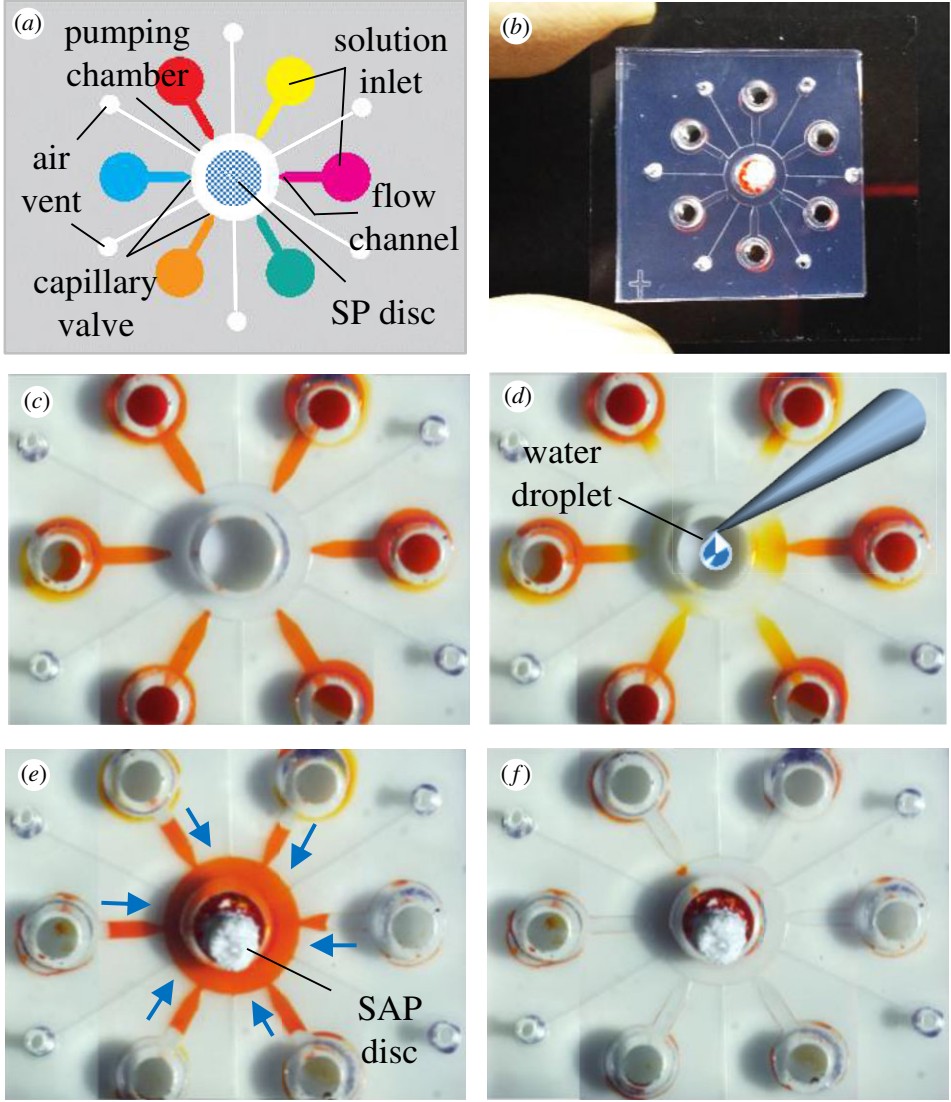

**Figure 4.** Multiple processing of solutions. (*a*) Structure of the device. (*b*) A photograph of the device. (*c*) Stopping of the solutions by the capillary valves. (*d*) Priming of a water droplet into the pumping chamber to allow solutions stopped at the valves to spread. (*e*) Pumping of multiple solutions. The blue arrows indicate the direction of solution flow. (*f*) The solutions are completely absorbed by a single polymer disc.

## 3.3. Solution processing using the micropump

Because of its simple structure and function, our micropump can easily be integrated into multiple-pump devices. Figure 4 shows a device with six solution reservoirs connected to a pumping chamber in the middle through flow channels with a width of 200 µm. In this device, the injection of the solutions into the pumping chamber was controlled using capillary valves (electronic supplementary material, figure S8), which can be used to suspend temporarily the solution flow [30]. Pumping is initiated by priming a water droplet into the pumping chamber [12], where the water spreads and merges with the solutions stopped at the valve region. Then, an SAP disc was placed in the pumping chamber to absorb the solution (electronic supplementary material, movie S3). Absorption proceeds until the reservoirs are empty.

To achieve multiplex processing of solutions, the valves should be able to open automatically. For this purpose, we developed in previous work a simple switchable hydrophobic valve (electronic supplementary material, figure S9) [28]. The valve consists of a hydrophobic SAM formed on a platinum electrode. The valve opens following the electrochemical desorption of the SAM from the electrode (electronic supplementary material). Figure 5 shows a test device with the pump, a reaction chamber, and a flow channel connected with the valves. The solution is injected into the reaction

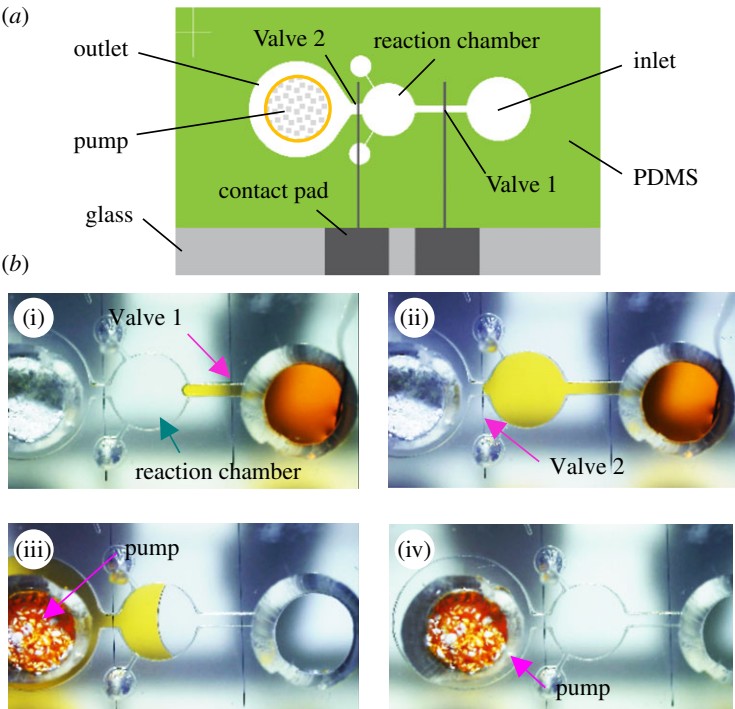

**Figure 5.** Controlled transport of a dye solution using the pump and the valves. (*a*) Layout of the device. (*b*) Operation of the pump and valves. (i) The solution is injected into the flow channel by opening Valve 1. (ii) The reaction chamber fills with the solution. (iii,iv) The solution is absorbed into the disc after Valve 2 opens.

chamber by opening the first valve (Valve 1). The second valve (Valve 2) suspends the solution flow. When Valve 2 is open, the solution flows into the pumping chamber and it is absorbed by the pump (electronic supplementary material, movie S4). Although we used an external potentiostat to apply the potential to the valve electrode, the valve can also be opened by shifting the potential of the electrode in combination with the other Zn/Pt electrode in a control flow channel [31].

As for automatic processing of multiple solutions, the use of liquid metal droplet as a surface tension mediator controlled by the application of electric signal [32] or ionic imbalance [33] may be possible. However, these pumps need an additional electrolyte solution. In this regard, passive solution transport based on capillary action and low-power switching realized by our pump will be more advantageous.

## 3.4. Application to chemical analysis

With the help of the microvalve, the micropump can be used to process solutions in various microfluidic systems. Electronic supplementary material, figure S10 shows a simple example for electrochemical detection of glucose. A three-electrode system was additionally formed along with the micropump and valves shown in figure 5 (electronic supplementary material, figure S10A). First, a glucose solution in the reservoir was injected into the detection chamber by opening the first valve (electronic supplementary material, figure S10B(i,ii)). After measurement, the solution was withdrawn by the micropump by opening the second valve (electronic supplementary material, figure S10B(iii,iv). The device produced a current that depended on glucose concentration (electronic supplementary material, figure S10C,D). More complicated processing of solutions, such as the exchange of solutions needed for affinity sensing, could be carried out by the combination of this simple procedure.

Electronic supplementary material, figure S11 shows a prospective scheme with the integration of the Pt-SAM valve and SAP pump to realize an automatic processing of many solutions required for bioassay. The valve system could perform the controlled and sequential injection of the component solutions in the assay site and the pump could remove the processed solutions from the site. Regarding this application, an experimental investigation is necessary.

# 4. Conclusion

In this study, we have described the design and practical realization of a simple micropump using SAP as an absorbing material. In our device, an SP disc is used to absorb solutions whose volumes are much larger than that of the dry disc. Despite its simple structure, this micropump can sustain a constant rate of absorption, which can be adjusted by changing the composition of the disc, the dimensions of the flow channel and the nature of the medium to be absorbed. Owing to its simple structure, our micropump is suitable for integration in multicomponent devices for multiple processing of solutions. The pumping can be initiated using a switchable hydrophobic valve, which could assist in the realization of automatic processing of many solutions for biochemical analysis. The simple micropump design we propose here may be useful for user-friendly disposable microfluidic devices with multiple solution processing functions.

Ethics. Human whole blood samples were obtained from Tsukuba i-Laboratory LLP. The Institutional Review Board at the University of Tsukuba reviewed and approved this study (no. 963).

Data accessibility. For the pump reproducibility, the data (including supporting movies) regarding the pump operational principle, characterization (e.g. absorbing rate, capacity), performance comparison with other material, control of pumping and applications have been presented in data file (data available from the Dryad Digital Repository: https://doi.org/10.5061/dryad.57k73c3 [34]).

Authors' contributions. G.C.B. and H.S. conceptualized the research idea and made the experimental design of the pump development, interpreted the data and prepared the manuscript. G.C.B. and M.M.R. fabricated the devices, conducted the experiments and processed the data. T.K. monitored the experiments with bio-samples including human blood and assisted in data interpretation. All the authors contributed to prepare the final manuscript and gave their final approval for publication.

Competing interests. There are no competing interests to declare.

Funding. This research work was partially supported by Grants-in-Aid for Scientific Research (no. 26560365) from Japan Society for the Promotion of Science (JSPS), Japan during the doctoral research work in 2015–2016 and research grant (Project code: LS/2017/04) of SUST research centre, Shahjalal University of Science and Technology, Bangladesh, respectively.

Acknowledgements. We thank Prof. Edwin T. Carlen, Mr Shishir Kanti Pramanik and Keisuke Tabata (University of Tsukuba) for their valuable discussions and assistance.

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
