## [Reviewer comments · Royal Society Open Science]

Review History

RSOS-182213.R0 (Original submission)

Review form: Reviewer 1

Is the manuscript scientifically sound in its present form?

Yes

Are the interpretations and conclusions justified by the results?

Yes

Is the language acceptable?

Yes

Is it clear how to access all supporting data?

No

Do you have any ethical concerns with this paper?

No

Have you any concerns about statistical analyses in this paper?

I do not feel qualified to assess the statistics

Recommendation?

Major revision is needed (please make suggestions in comments)

Comments to the Author(s)

The paper is well written and the new developed pump is very well described. However, it is highly recommended to:

-make a table at the discussion section to compare your achievements with what you presented in the literature as being shortcomings of current products.

-also, every pump needs good reliability. A discussion about the life of the pump and any efficiency changes during the life time will help the reader highly.

Review form: Reviewer 2

Is the manuscript scientifically sound in its present form?

Yes

Are the interpretations and conclusions justified by the results?

Yes

Is the language acceptable?

Yes

Is it clear how to access all supporting data?

Not Applicable

Do you have any ethical concerns with this paper?

No

Have you any concerns about statistical analyses in this paper?

No

Recommendation?

Accept with minor revision (please list in comments)

Comments to the Author(s)

The authors presented a simple yet innovative micropump. This is a nice idea for creating disposable microfluidics. They developed the idea based on a superabsorbent polymer freeze-dried material. The material absorbed the liquid in the flow channel and initiates pumping by swelling. They showed that the rate of absorption remained relatively constant and was a function of the characteristic of the absorbent material. They also presented the electrical signal response for switching valve that turns on the pump.

I support the publication of this work with just a few minor comments:

1- That the authors mention liquid metal based pumps which are also very low cost (PNAS 111, 3304, 2014, and Nature Comm 7, 12402, 2016)

2- I strongly suggest that, if it is allowed by the journal, the authors include several of the videos that show the performance of the pump in action

Decision letter (RSOS-182213.R0)

31-Jan-2019

Dear Dr Biswas,

The editors assigned to your paper ("A simple micropump based on a freeze-dried superabsorbent polymer for multiplex solution processing in disposable devices") have now received comments from reviewers. We would like you to revise your paper in accordance with the referee and Associate Editor suggestions which can be found below (not including confidential reports to the Editor). Please note this decision does not guarantee eventual acceptance.

Please submit a copy of your revised paper before 23-Feb-2019. Please note that the revision deadline will expire at 00.00am on this date. If we do not hear from you within this time then it will be assumed that the paper has been withdrawn. In exceptional circumstances, extensions may be possible if agreed with the Editorial Office in advance. We do not allow multiple rounds of revision so we urge you to make every effort to fully address all of the comments at this stage. If deemed necessary by the Editors, your manuscript will be sent back to one or more of the original reviewers for assessment. If the original reviewers are not available, we may invite new reviewers.

- Data accessibility

It is a condition of publication that all supporting data are made available either as supplementary information or preferably in a suitable permanent repository. The data accessibility section should state where the article's supporting data can be accessed. This section

should also include details, where possible of where to access other relevant research materials such as statistical tools, protocols, software etc can be accessed. If the data have been deposited in an external repository this section should list the database, accession number and link to the DOI for all data from the article that have been made publicly available. Data sets that have been deposited in an external repository and have a DOI should also be appropriately cited in the manuscript and included in the reference list.

If you wish to submit your supporting data or code to Dryad (<http://datadryad.org/>), or modify your current submission to dryad, please use the following link:
<http://datadryad.org/submit?journalID=RSOS&manu=RSOS-182213>

- **Competing interests**

- **Authors' contributions**

- **Acknowledgements**

- **Funding statement**

Kind regards,

Royal Society Open Science Editorial Office
Royal Society Open Science
openscience@royalsociety.org

on behalf of Dr Derek Abbott (Associate Editor) and Professor R. Kerry Rowe (Subject Editor)
openscience@royalsociety.org

Comments to Author:

Reviewers' Comments to Author:

Reviewer: 1

Comments to the Author(s)

The paper is well written and the new developed pump is very well described. However, it is highly recommended to:

-make a table at the discussion section to compare your achievements with what you presented in the literature as being shortcomings of current products.

-also, every pump needs good reliability. A discussion about the life of the pump and any efficiency changes during the life time will help the reader highly.

Reviewer: 2

Comments to the Author(s)

The authors presented a simple yet innovative micropump. This is a nice idea for creating disposable microfluidics. They developed the idea based on a superabsorbent polymer freeze-dried material. The material absorbed the liquid in the flow channel and initiates pumping by swelling. They showed that the rate of absorption remained relatively constant and was a function of the characteristic of the absorbent material. They also presented the electrical signal response for switching valve that turns on the pump.

I support the publication of this work with just a few minor comments:

1- That the authors mention liquid metal based pumps which are also very low cost (PNAS 111, 3304, 2014, and Nature Comm 7, 12402, 2016)

2- I strongly suggest that, if it is allowed by the journal, the authors include several of the videos that show the performance of the pump in action

Author's Response to Decision Letter for (RSOS-182213.R0)

See Appendix A.

Decision letter (RSOS-182213.R1)

27-Feb-2019

Dear Dr Biswas,

I am pleased to inform you that your manuscript entitled "A simple micropump based on a freeze-dried superabsorbent polymer for multiplex solution processing in disposable devices" is now accepted for publication in Royal Society Open Science.

You can expect to receive a proof of your article in the near future. Please contact the editorial office (openscience_proofs@royalsociety.org and openscience@royalsociety.org) to let us know if

you are likely to be away from e-mail contact. Due to rapid publication and an extremely tight schedule, if comments are not received, your paper may experience a delay in publication.

on behalf of Dr Derek Abbott (Associate Editor) and Professor R. Kerry Rowe (Subject Editor)
openscience@royalsociety.org

Appendix A

Response to the referees

We really appreciate the referees who reviewed our manuscript carefully spending their precious time for us. The points raised by the referees have been checked carefully and the corresponding portions have been revised. The manuscript file with revised portions highlighted in blue (and underlined) has been uploaded as Supporting Information for Review Only.

Referee: 1

The paper is well written and the new developed pump is very well described.

→ We thank the referee for his/her comment.

However, it is highly recommended to:

-make a table at the discussion section to compare your achievements with what you presented in the literature as being shortcomings of current products.

→ We thank the referee for his/her advice. We have added a comment at the corresponding portion (page 4, section 4.2, line 42-43) and added a table with information regarding the problems with current passive pumps and improvements realized by our pump (page 4, at the end of section 4.2).

-also, every pump needs good reliability. A discussion about the life of the pump and any efficiency changes during the life time will help the reader highly.

→ We thank the referee for his/her comment. An excellent property of our pump is its stability for long-term storage. The SAP discs could be stored for more than one month in an air-tight container maintaining the initial performance as freshly prepared ones. On the other hand, A_c and A_r decreased by approximately 10% when the discs were stored in the open-air condition at room temperature for 30 days. Even in this case, the deterioration of the performance was not significant. The result indicates

that the SAP pumps can be used reproducibly if they are stored properly. We have added a comment (page 4, section 4.2, line 37-41).

Referee: 2

Comments to the Author(s):

The authors presented a simple yet innovative micropump. This is a nice idea for creating disposable microfluidics. They developed the idea based on a superabsorbent polymerfreeze-dried material. The material absorbed the liquid in the flow channel and initiates pumping by swelling. They showed that the rate of absorption remained relatively constant and was a function of the characteristic of the absorbent material. The also presented the electrical signal response for switching valve that turns on the pump.

I support the publication of this work with just a few minor comments:

→ We thank the referee for his/her careful evaluation and kind consideration for publication.

1- That the authors mention liquid metal based pumps which are also very low cost (PNAS 111, 3304, 2014, and Nature Comm 7, 12402, 2016)

→ We thank the referee for his/her advice. We have additionally cited those two papers in the introduction with comments. (page 5, section 4.3, line 19-22).

2- I strongly suggest that, if it is allowed by the journal, the authors include several of the videos that show the performance of the pump in action

→ We thank the referee for his/her advice. We have added four movies to show the operation of our pump. The information about the movies has been added as follows

Supporting movie 1 → Section 3.5, page 3, line 29

Supporting movie 2 → Section 4.2, page 4, line 27

Supporting movie 3 → Section 4.3, page 5, line 8

Supporting movie 4 → Section 4.3, page 5, line 16